# Influence of Living and Dead Roots of Gansu Poplar on Water Infiltration and Distribution in Soil

**Dashuai Zhang** [1,2]**, Yao Dai** [1,2]**, Lingli Wang** [3,4] **and Liang Chen** [1,2,]*

1   State Key Laboratory of Hydraulic Engineering Simulation and Safety, Tianjin University, Tianjin 300072, China; zhangdashuai1014@163.com (D.Z.); 15222639739@163.com (Y.D.)
2   School of Civil Engineering, Tianjin University, Tianjin 300072, China
3   School of Environmental Science and Engineering, Tianjin University, Tianjin 300072, China; wanglingli@sinomine.cn
4   Sinomine Rock and Mineral Analysis (Tianjin) Co. Ltd., Tianjin 300270, China
*   Correspondence: liangchen@tju.edu.cn

**Abstract:** During rapid urbanization, it is necessary to increase soil permeability and soil porosity for reducing urban runoff and waterlogging risk. Woody plants are known to increase soil porosity and preferential flow in soil via living roots growth and dead roots decay. However, the primary results of dead woody plant roots on soil porosity and permeability have been discussed based only on the hypotheses or assumptions of different researchers. In this study, living and dead roots (decayed under natural conditions for more than 5 years) of Gansu poplar trees (*Populus gansuensis*) were selected. They were selected to compare the influence between living and dead roots on water infiltration rate and soil porosity in a cylindrical container (diameter = 20 cm, height = 66 cm) under laboratory conditions. Results indicated that the steady-state water fluxes at the bottom of the containers without roots (control), with living roots, and with dead roots were 54.75 ± 0.80, 61.31 ± 0.61, and 55.97 ± 0.59 cm d$^{-1}$, respectively. Both living roots and dead roots increased the water infiltration rates in soil and also increased the water storage capacity of soil. The water storage capacities of soil without roots, with living roots, and with dead roots were 0.279, 0.317, and 0.322 cm$^3$ cm$^{-3}$, respectively. The results from SEM indicated that smaller pores (30–50 μm) were in living roots and larger pores (100–1000 μm) were in dead roots. The soil permeability was increased by living roots possibly due to the larger channels generated on the surface of the roots; however, water absorbed into the dead roots resulted in greater water storage capacity.

**Keywords:** soil permeability; soil porosity; water storage capacity; woody plant; desert poplar; *Populus gansuensis*

## 1. Introduction

The infiltration rate of underlying surfaces quickly decreases during rapid urbanization due to the dramatic increase in impervious land area. This has led to increases in urban runoff and waterlogging risk. Woody plants have been proven as a new effective tool in urban stormwater management because woody plants can increase soil porosity and preferential flow by living roots growth and dead roots decay [1–3].

Thompson et al. [4] investigated the biomass–infiltration relationships from nearly 50 vegetation communities that spanned from hyperarid deserts to the humid tropics (representing a full spectrum of soil types). They found that soil permeability increased as a power law function of aboveground biomass in water-limited ecosystems. Bartens et al. [5] observed that roots of Black oak (*Quercus velutina Lam.*) and red maple (*Acer rubrum L.*) trees could penetrate the geotextile and subsoil. In structural

soil profile study, trees increased the average infiltration rate 27-fold compared to unplanted controls. In a greenhouse study, trees increased infiltration rate by an average of 153%. Mishra and Sharma [6] also reported that the soil porosity and soil water holding capacity increased from 41.2% to 46.3% and from 4.3 to 4.8 g kg$^{-1}$ with the growth of *Prosopis juliflora* (a woody plant) within 3 years. They also reported that the bulk density of surface soil decreased from 1.66 to 1.37 t m$^{-3}$. With the growth of *Malus baccata* with fibrous roots and *Sophora japonica* with tap roots, Zhang et al. [7] observed that water infiltration rates increased at ratios of 19% and 118%, respectively. A larger increase of infiltration rate by *S. japonica* was observed due to its deeper and more vertical tap roots.

For living roots, the growth of roots can split soil into larger pores and increase soil porosity by 30% [1,3]. These larger pores generated by living roots can let more air enter into the soil and promote animals breathing and activities. The soil pores are key factors affecting water infiltration and hydraulic conductivity in soil by controlling the resistance against gravity and suction [8,9]. Gyssels and Poesen [10] and De Baets et al. [11] found that soil porosity had a positive influence ($R^2 > 0.9$) on infiltration rates, and soil pore distributions were mainly affected by roots distribution [12–14]. Many relationships between the distribution of different pore sizes and hydraulic conductivities have been derived by different researchers including Childs and Collis-George [15], Marshall [16], Nielsen et al. [17], and Jackson et al. [18]. Moreover, during the growth of roots, roots and microorganisms in the rhizosphere can increase organic matter in soil [19] and produce hyphae and exudate [20,21]. They could positively affect soil pores and soil structure by releasing organic compounds and combining and aggregating soil particles [22]. Jiménez et al. [23] found that the green forest which had the highest organic matter had the highest infiltration rates, and *Spartocytisus* which had the lowest organic matter had the lowest infiltration rates.

For dead roots, Sun et al. [24] reported that only 35% of initial mass, on average, of 35 woody plant roots decayed after 6 years of exposure in the field. Root channels and preferential flow channels are formed in place by decaying roots [25,26]. After roots decay, many spaces were left to become macropores [27,28], which increased pore air volume, increased water infiltration rate, and reduced soil interaction [29]. Mitchell et al. [25] reported that dead alfalfa with taproot growth could produce stable macropores and improve water infiltration rates in soil; however, dead wheat with fibrous roots had no significant influence on water infiltration rates in soil. Moreover, during the decay of woody plant roots, organic matter also is generated and can improve soil porosity by altering soil structure [30,31]. Shortle and Dudzik [32] summarized the patterns of wood decay and observations of thousands of trees and reported that after the younger or smaller trees died, they were decayed to increase the organic matter and nutrients, which could increase soil porosities and decrease the soil bulk densities [33].

Similar to living roots, dead roots may also increase soil porosity. However, no previous studies have been conducted to investigate the influence of dead roots on soil porosity and soil permeability nor to compare their effects on water infiltration rate and water storage capacity of soil. In this study, we hypothesized that the pores created by dead roots would increase water infiltration and water storage more than living roots, as the void space from dead roots can serve as macropores. Living and dead roots of Gansu poplar (*Populus gansuensis*) were selected because this species is widely planted in northern China and it has waterlogging and salt tolerance [34–36]. Batch-type tests were conducted to compare the accelerating influence between living and dead roots on water infiltration rate and water storage capacity of soil via the volumetric method and water balance method in the laboratory.

## 2. Materials and Methods

### 2.1. Experimental Design and Materials

The test was conducted in a cylindrical container made of Plexiglas (diameter = 20 cm, height = 66 cm, Figure 1) under laboratory conditions. The container was filled with gravel at the bottom, soil in the middle, and gravel at the top. The top, middle, and bottom layer thicknesses



were 10, 36, and 10 cm, respectively. The upper gravel layers were designed to ensure the water infiltrated uniformly. The lower gravel layers allowed water to drain freely during measurements. Three layers of nonwoven cloth were placed inside the container between the gravel and soil to prevent soil loss during water percolation. A constant head control outlet was provided at a 5 cm distance from the gravel surface to maintain a constant head. A lower bottle (volume = 5,000 mL) was placed above the container to maintain a constant head of water on the soil surface.

The soil used was collected from green land in the Peiyang campus of Tianjin University, and the soil type was silty clay. The soil was screened through a 20 mesh sieve before being added to the cylindrical container. In order to obtain same bulk density in each cylindrical container, soil samples with same weight were added to each cylinder at the same height several times. The soil density in each cylindrical container was 1.21 g cm$^{-3}$.

The living and dead roots of the poplar tree species were added to the middle of the cylindrical container during the addition of soil. A cylindrical container without roots was also set as the blank test. The living and dead roots of Gansu poplar trees (*Populus gansuensis*) were collected from Gansu province, China, with densities at 0.6 and 0.5 g cm$^{-3}$, respectively. The dead roots had been decaying for more than 5 years under natural conditions.

The experiment was conducted in March 2018. The water supply time and experiment duration were 21 and 30 h, respectively. The water used was tap water obtained from the Peiyang campus of Tianjin University. The water quality analysis results for the tap water are shown in Table 1.

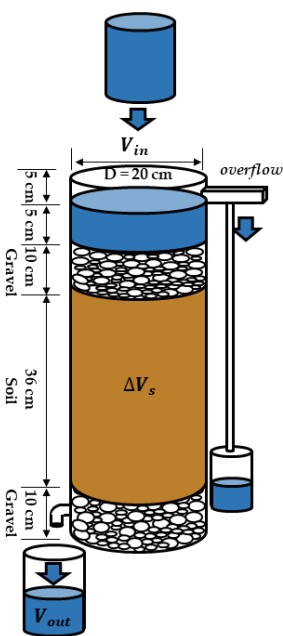

**Figure 1.** A conceptual diagram of the experiment.

**Table 1.** Water quality analysis results for tap water.

| pH | TDS | TOC | Cl$^-$ | SO$_4{}^{2-}$ | HCO$_3{}^-$ | CO$_3{}^{2-}$ | Ca$^{2+}$ | K$^+$ | Mg$^{2+}$ | Na$^+$ |
|---|---|---|---|---|---|---|---|---|---|---|
| (-) | ($\mu$S cm$^{-1}$) | (mg L$^{-1}$) | (mg L$^{-1}$) | (mg L$^{-1}$) | (mg L$^{-1}$) | (mg L$^{-1}$) | (mg L$^{-1}$) | (mg L$^{-1}$) | (mg L$^{-1}$) | (mg L$^{-1}$) |
| 8.02 | 284 | 2.40 | 9.94 | 24.90 | 114 | <5 | 39.0 | 1.98 | 7.77 | 8.16 |

## 2.2. Volumes of Flow and Water Flux at the Bottom Outlet

Under the condition of constant water flow ($Q$ = 45,714 mL d$^{-1}$), the soil outflow rate at the bottom outlets ($Q_{out}$, mL d$^{-1}$) was measured via the volumetric method (500 mL container) and recorded every twenty minutes [7]. The penetration time of water ($T$, min) at the bottom was also recorded. All containers were weighed at the beginning and end of the experiment to calculate the volume

of water storage ($\Delta V_s$, mL) of soil without or with the living and dead roots of poplar. During the experiment, the influence of evaporation was ignored as the experiment time was short. The soil accumulated inflow at the top of the container ($V_{in}$, mL), the soil accumulated outflow at the bottom outlets ($V_{out}$, mL), and water flux at the bottom outlet ($I_{out}$, cm d$^{-1}$) were calculated by water balance equations as below [7,37]:

$$V_{in} = V_{out} + \Delta V_s \tag{1}$$

$$V_{out} = \Sigma\, Q_{out} \tag{2}$$

$$I_{out} = Q_{out}/A \tag{3}$$

where $A$ is the area of soil inside of the container ($A = 314$ cm$^2$).

### 2.3. SEM

Based on the hypotheses of different researchers [1,2,22], the different effects may not only be due to the different influence of living and dead roots on soil pores but also due to the different pore sizes in living and dead roots. Therefore, the pores of living and dead roots were examined by a scanning electron microscope (SEM). The living and dead roots were kept in soil for several days and dried naturally in Tianjin University Laboratory after the tests. Then, they were cut horizontally into 1.5 mm slices. In order to make the roots conductive, the slices were fixed on slides and sprayed with copper for 40 min. A Hitachi S-4800 scanning electron microscope (Japan) was used to test the samples, and the images were scaled to 1 mm and 50 μm.

## 3. Results and Discussion

### 3.1. Volumes of Water Distribution in Soil

Volumes of water distribution (the accumulated inflow at the top of container/$V_{in}$, the accumulated outflow at the bottom of container/$V_{out}$, and the volume of water storage/$\Delta V_s$) without or with living and dead roots of poplar are shown in Figure 2. Compared to the container without roots ($V_{in}$ is 19,198 mL, $V_{out}$ is 16,046 mL, and $\Delta V_s$ is 3,152 mL), the accumulated inflows at the top of the container with living and dead roots increased by 21.4% and 17.3%, respectively. The accumulated outflow at the bottom of container with living and dead roots increased by 22.9% and 17.7%, respectively. The volume of water storage of soil in containers with living and dead roots increased by 13.7% and 15.4%, respectively. Living and dead roots can significantly increase the infiltration rate of soil and improve the water storage capacity of soil. However, a larger effect of living roots on the infiltration rate and a larger effect of dead roots on water storage capacity were observed.

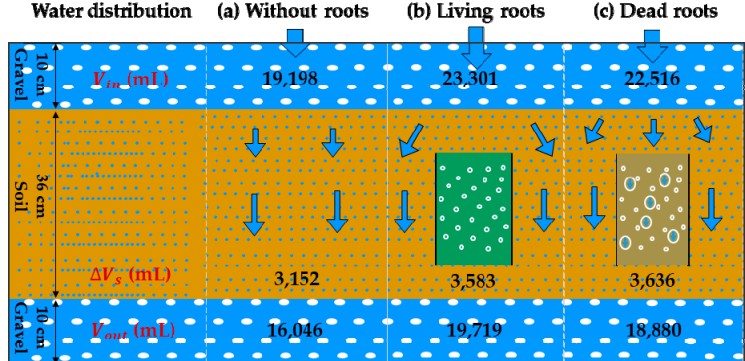

**Figure 2.** Water distribution and water flow in soil without or with living and dead roots.

### 3.2. Temporal Variation in Water Flux at the Bottom Outlets

The temporal variations of water flux at the bottom of containers ($I_{out}$) without or with living and dead roots of poplar were calculated and shown in Figure 3. This was related to soil permeability and groundwater recharge [38,39]. Similar trends of $I_{out}$ in containers without or with living and dead roots were observed and divided into three stages. Compared to the container without roots, the existence of living and dead roots can accelerate the water movement in soil.

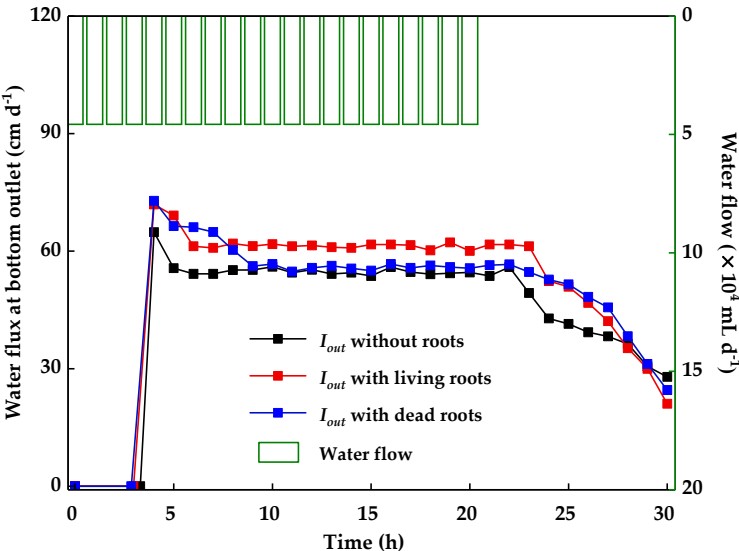

**Figure 3.** Water flux at the bottom of containers without or with living and dead roots.

In the first stage (0–9 h), the soil was unsaturated, and the soil pores were not filled with water ($I_{out}$ increased). The value of $I_{out}$ at the beginning was 0 cm d$^{-1}$ and increased with the continued infiltration of water. The time for the first drop of water to flow out of the container bottom was recorded and shown in Table 2. Compared to the container without roots, the existence of living and dead roots accelerated the water movement in soil. Also, a larger effect of dead roots on the time of the first drop of water flow out of the bottom was observed.

In the second stage (9–21 h), the soil was saturated, and the flux at the top the container was balanced with the water flux at bottom outlet ($I_{out}$ stabilized). The steady-state water fluxes at the bottom of the container were calculated and are shown in Table 2. Compared to the container without roots, the steady-state water flux at the bottom of containers with living roots and dead roots increased by 12.0% and 2.2%, respectively. The existence of living and dead roots accelerated the water infiltration in soil, and a larger effect of living roots on the steady-state water flux was observed. The effect of living roots on soil permeability was similar to the effect observed by Zhang et al. [7], where they reported that *M. baccata* with fibrous roots could increase infiltration rate by 19%. Bartens et al. [5] reported that black oak and red maple could increase the infiltration rate by an average of 153% in their greenhouse study. The different impact levels on soil permeability are related to the different soil densities, soil pores, and root conditions in each test [8,12–14]. The observed effect of dead roots on soil permeability was similar to the effect observed by Mitchell et al. [25], where they reported that dead alfalfa with taproot growth could improve water infiltration rates in soil. Many spaces were left to become macropores after root decay, which could increase the water infiltration rate [25,27,28]. This finding provided stronger evidence for the effect of dead roots on soil permeability.

In the third stage (21–30 h), water flow was terminated, and the soil became unsaturated again ($I_{out}$ decreased). When the water flux at the bottom of the container was terminated (0 cm d$^{-1}$), some water was still stored in soil. The water storage capacities of soil were calculated according to Figure 2 and are shown in Table 2. Compared to the container without roots, the water storage capacities of soil with living roots and dead roots increased by 13.7% and 15.4%, respectively. The effect of living roots on the

soil water storage capacity was similar to the effect observed by Mishra and Sharma [6], where they reported that plant roots could increase the soil water storage capacity by 11.6%. Living roots increased soil pores [1,3,12–14] and could store water in soil. Additionally, the results of dead roots about water storage capacity of soil agree with the hypothesis that dead roots could increase water storage capacity of soil, which contribute to the macropores left by dead roots.

**Table 2.** Penetration time, steady-state water flux at the bottom of container, and water storage capacity of soil without or with living and dead roots.

| Test Design | Without Roots | Living Roots | Dead Roots |
|:---:|:---:|:---:|:---:|
| Penetration time (min) | 198 | 178 | 171 |
| Steady-state water flux at the bottom of the container (cm d$^{-1}$) | 54.75 ± 0.80 | 61.31 ± 0.61 | 55.97 ± 0.59 |
| Water storage capacity of soil (cm$^3$ cm$^{-3}$) | 0.279 | 0.317 | 0.322 |

### 3.3. Electron Micrographs of Root Macropores

According to the results above, living roots accelerated the water infiltration rate in soil, but dead roots further increased water storage capacity of soil. These different effects may not only be due to the different influence of living and dead roots on soil pores but also due to the different pore sizes in living and dead roots. Other studies have been conducted on the influence of living and dead roots on soil pores [1,3], soil permeability [5–8], and soil water storage capacity [25], but no study is known to have investigated the influence of pore sizes in living and dead roots. To our knowledge, this is the first time the pores of living and dead roots were examined via a Hitachi S-4800 electron microscope in this study. The smaller pores with diameters from 30–50 μm in living roots were observed (Figure 4a,b), and the larger pores with diameters from 100–1,000 μm in dead roots were observed (Figure 4c,d).

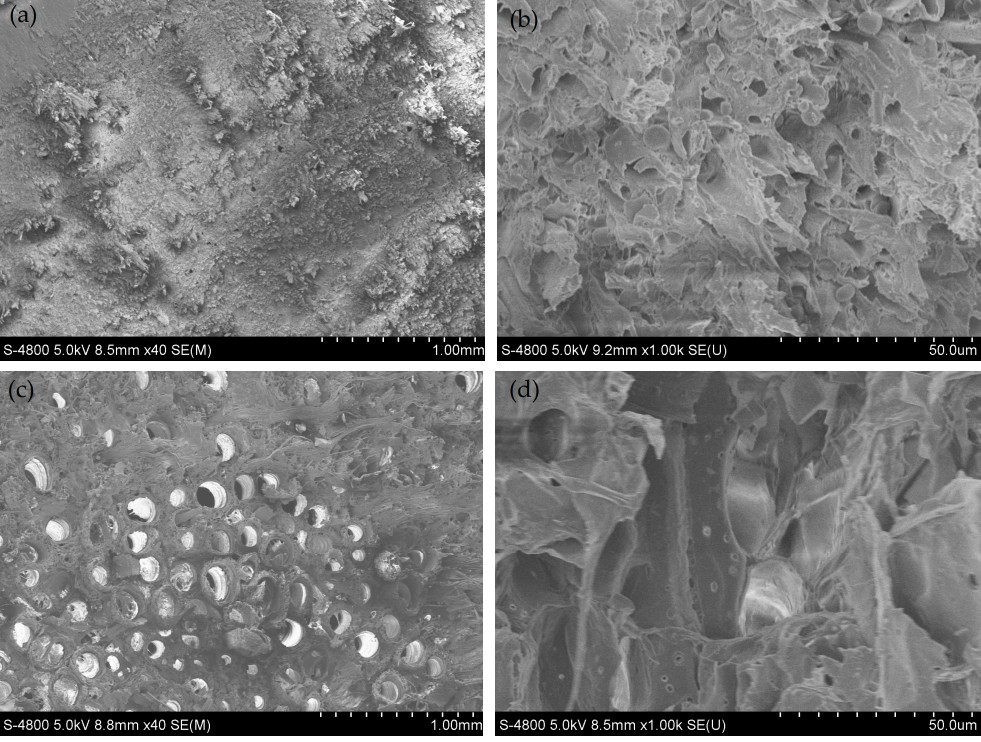

**Figure 4.** Electron micrographs (via SEM) of root macropores: (**a,b**) internal pores of living roots (scale bars: 1 mm and 50 μm, respectively); (**c,d**) internal pores of dead roots (scale bars: 1 mm and 50 μm, respectively).

As summarized in Figure 5, living roots had relatively smaller pores with diameters of about 30–50 μm, and they had a much greater water infiltration rate in soil (61.31 cm d$^{-1}$) than dead roots (55.97 cm d$^{-1}$) and without roots (control, 54.75 cm d$^{-1}$). Dead roots had relatively larger pores with diameters of about 100–1,000 μm, and they had a much greater water storage capacity of soil (0.322 cm$^3$ cm$^{-3}$) than living roots (0.317 cm$^3$ cm$^{-3}$) and without roots (control, 0.279 cm$^3$ cm$^{-3}$). Combining the different influences of living and dead roots on the water storage capacity of soil and on the water infiltration rate in soil, we speculate that soil permeability improved by living roots may be due to the dominant channel generated on the roots surface. However, water could penetrate into the dead roots, and more water was stored by the pores in the dead roots.

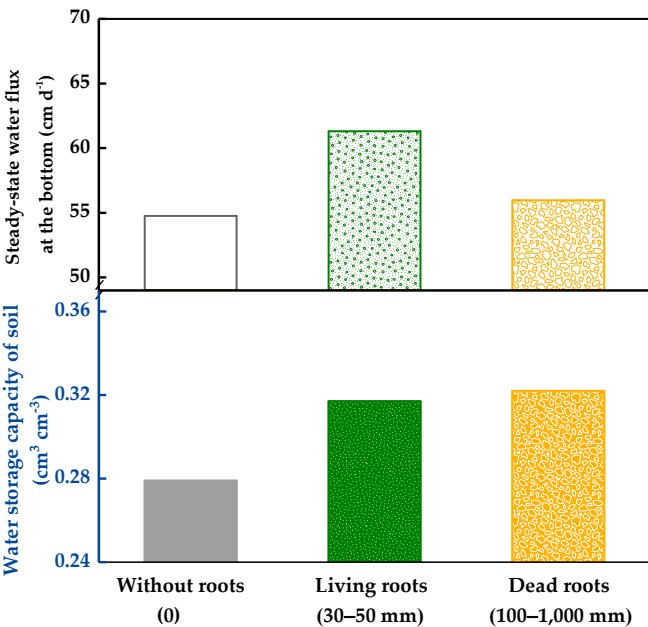

**Figure 5.** The steady-state water flux at the bottom of containers, water storage capacity of soil, and the diameter of internal pores without or with living and dead roots (soil density is 1.21 g cm$^{-3}$).

## 4. Conclusions

Batch-type experiments were conducted to investigate the influence of living and dead roots (decayed under natural conditions for more than 5 years) of poplar (a woody plant species) on water infiltration rate and water storage capacity of soil. Results showed that living and dead roots could increase the infiltration rate in soil and improve the water storage capacity of soil. However, a larger effect of living roots on the infiltration rate and a larger effect of dead roots on the water storage capacity of soil were observed. In addition, smaller pores (30–50 μm) were observed in living roots and larger pores (100–1,000 μm) were observed in dead roots. Soil permeability was improved by living roots due to the dominant channel generated on the roots surface; however, water could penetrate into the dead roots, and more water was stored by the pores in the dead roots.

**Author Contributions:** Writing—original draft preparation, D.Z.; Conducting the experiment, D.Z., Y.D., and L.W.; Writing—review and editing, L.C.; Funding acquisition, L.C. All authors have read and agreed to the published version of the manuscript.

**Funding:** This work was supported by the National Key R&D Program of China (2016YFC0401405), the National Natural Science Foundation of China (No. 41772245), and the Science Fund for Creative Research Groups of the National Natural Science Foundation of China (No. 51621092).

**Acknowledgments:** The English in the manuscript was edited by Springer Nature Author Services.

**Conflicts of Interest:** The authors declare no conflict of interest.

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
