# Peer review of "Influence of Living and Dead Roots of Gansu Poplar on Water Infiltration and Distribution in Soil"

_applsci, doi:10.3390/app10103593_

Round 1

Reviewer 1 Report

This is a revised manuscript of the paper that I was already reviewed the original version of it. The revised manuscript is more readable, organized, and written well compared to the first version. In the first version, the author tested two kind of soil (two groups). In the revised manuscript, the author tested only one kind of the soil. Could the authors state why they removed the other group of soils?

The authors addressed well some of my comments on first version. However, the authors did not address other comments that I mentioned in the first version and listed below:

  • Lines 78: Why the authors used this kind of plant (poplar)? What is the benefit of this plant? This is should be clearly stated in the introduction before presented the objective of the study.
  • Lines 93-97: The percentage of sand, silt, and clay of soils should be stated.       Did the authors use more than one soil type? The soils should be analyzed according to ASTM standards
  • Figure 1: Point to the location of gravels and soil sample.
  • Lines 119-120: Please, add reference to Eqs. 1 through 3.
  • Lines 149-159: The authors stated that “In the first stage (0 -9h), the soil was not stable (Ioutincreased).”, “In the second stage(9-21h), the soil was stable (Ioutstabilized)” and “In the third stage (21 -30 h), the inflow water at the top was cut and the soil became unstable again (Ioutdecreased)”. They used the term “stable” and “unstable” instead of saturated and not saturated. Are there any previous studies used these terms? This is should be clarified by the authors and discussed these terms with previous studies.

Author Response

Dear Reviewer,

Please see the attachedment. Thanks.

Reviewer 2 Report

Dear Authors, this paper is qualitative work. It needs some improves. I know that after minor correction of paper it will be better. In attached file you can found paper with comments.

Kind regards, reviewer.

Author Response

(The authors gave the same response as above.)

Reviewer 3 Report

General comments:

The study by Zhang et al investigated with live or dead roots increased the flow rate and porosity of soils under laboratory conditions of columns. Although the quality of the writing is very good, the did not do a good job using existing literature to compare their results and their introduction needs to be strengthened as described below before it should be accepted.

The purpose of highlighting the text different colors was unclear and should be removed.

The introduction is missing a clearly described objective and set of hypotheses tested. Author need to clearly state the hypotheses such as  “We hypothesized that the pores created by dead roots would increase water infiltration and water storage greater than living roots and controls as void space from dead roots can serve as macropores.” Moreover, the authors need to clearly state their means of experimentation in the introduction. What are the assumptions, advantages, and drawback of using laboratory columns to experiment?

The results and discussion section needs to utilize existing literature to see if the increase in flow rate is similar, greater, or lower than in the literature. Moreover, the authors need to justify the use of SEM for small porosity as the introduction made the tap root and larger roots seem most important. Again, the authors need to use the literature to help actually discuss their results and interpret their values. I think additional figures comparing the control, live roots, and dead roots porosity, flow rates, and soil water holding capacity are needed.

Specific comments:

Abstract:

Rephrase the first sentence to focus on the problem. Why is soil porosity and its measurement/formation important?

Line 13: Change ‘nature’ to ‘natural’.

Line 13: Briefly describe where and the conditions this study were conducted. Are these urban forests, laboratory conditions, or in developed, metropolitan areas?

Line 17: What are the units for water storage? Is this per kilogram of soil? Per soil pot? These units are very unclear.

Line 19: What are the columns? Please describe them briefly in the abstract.

Line 21: Briefly describe how soils were collected to measure porosity? Are these aggregates and were soils dried before making sections of SEM?

Introduction:

Line 29: Change ‘had been proved’ to ‘have been proven’.

Line 29: I believe the authors should start with the overarching problem that poor water infiltration is bad because it increases runoff to during storms, etc.

Line 37-41, please rephrase the sentence so that units and values are next to their variable.

Line 45: rephrase this sentence as it seems contradictory: large pores are decreased but soil porosity increases.

Line 51: Change ‘R2>0.9’ to ‘R> 0.9’. by adding spaces.

Line 58: change ‘rearranging’ to the more commonly used term  ‘aggregating’.

Line 62: What level of decay occurred over 6 years? Roots can have decay but largely be in tact physically.

Methods:

Line 106: Authors should briefly describe the quality of the water (EC, pH, TDS)

Line 122: Where samples dried or kept moist? Authors should briefly state the importance of maintain moisture with respect to preventing formation of new porosity during drying.

Results and Discussion section

Figure 2: This diagram is unclear, is this vertical or horizontal? Please remake.

Section 3.2: How do these results compare to previous studies conducted on this topic?

Line 147: This line is unclear, did the roots accelerate water movement or not? This section needs to be re-written with more clarity and exact language.

Section 3.3: Authors need to describe assumptions and issues with the methodology. Also, why look at such small pores?

Author Response

(The authors gave the same response as above.)

Round 2

Reviewer 1 Report

The authors are addressed well all the comments that raised by the reviewers. I think that the manuscript is good to go publishing.  

Author Response

Dear Reviewer,

        Thanks for your meaningful comments.

Reviewer 3 Report

The authors have done a good job answering many of my correction, comments, and concerns about the manuscript. I appreciate their revisions to their figures and adding information critical to future readers.

The authors, however, did not do an adequate job with revising their results and discussion sections. still do not put their research findings into context with previous studies and what their results mean for their problem/question posed in the introduction.

The most clearest example, my comment #3 has not been adequately addressed as the study still does not 'utilize existing literature to see if the increased flow rate is similar to values reported' and 'justify the use of SEM and interpret their results and values'.

Values are not stated and the SEM images still serve as paintings on a wall as they are not interpreted nor compared with the size expected and reported in other studies.

I leave it up to the editor if they wish to request the authors to comply and add these requested major revisions or accept pending minor additions to comment #3.    

Author Response

(The authors gave the same response as above.)

Round 3

Reviewer 3 Report

I thank the authors for making the additions to better interpret their SEM images and include additional references. I believe the study is ready to be accepted.

Author Response

Dear Reviewer,

Thanks for your meaningful comments.

This manuscript is a resubmission of an earlier submission. The following is a list of the peer review reports and author responses from that submission.

Round 1

Reviewer 1 Report

Unfortunately, this manuscript suffers from multiple fundamental issues that prevents it from publication. The issues are linked to the fundamental misunderstanding of soil structure and hydrology in interaction with root system. In lines 88-91, authors mentioned that they have added gravel to the top and bottom of the soil column to get a uniform flow. First, water flow in in porous medium is almost always one directional, from macropores to micropores. Putting a layer of “gravel” at the bottom of soil column can completely change the water drainage at the transition interface and stop the water percolation until the hydrostatic pressure or height of perched water on top force it through. Second, authors stated that by doing so they were trying to uniform the water flow while they were studying the irregularity of flow due to macropores created by root system. Third, in line 98-99, authors mentioned that they have sieved the collected soil sample before locating within cylinder. Sieving soil disturbs its natural structure and loose its connection with the naturally evolved soil system. They even further modified the water flow and soil porous system by locating one of three “unwoven cloth” (line 91) at the center of section with soil sample. It is not clear how roots are supposed to pass through these screens. Fourth, in lines 105-106, authors explain that they have added “dead and lining” roots to the mid-section. Root pieces that are artificially added to soil are completely different from ones that have been grown in place. Interaction of root system with soil microorganisms and its effect on soil structural dynamics cannot be lowered to the physically formed tube-like channels.  Fifth, the schematic view of water flux in relationship with root as illustrated in Figure 3 shows the clear bias that authors have in understanding of root system and soil hydrology. They have lowered the complex physiological process of living roots interaction with water; the effect of energy status on water transmission through semi-permeable membranes, the huge difference in size and characteristics of soil and membrane pores to tube-like channels that can direct water both from cut section and walls.  Moreover, there are many examples of odd phasing and unclear statements as exemplified in the following specific comments. Most importantly, authors have not provided and clear objective that is meaningfully linked to the natural environment.   

Specific comments

Abstract

Lines 16-22: the wording here is extremely confusing. Please consider revising the entire section to clearly differentiate the flow characteristics for tests with and without roots, dead and living roots, and first and second group.

Introduction

Lines 38: roots generally create macropores by modifying pore size distribution.

Line 41: replace “concerning” with “affecting”.

Line 42: Is the reported R2 value associated with porosity or infiltration?

Line 46: please either use italic font for all specific names or none of them “Black oak” has been written differently in lines 47 and 47.

Lines 50-51: change the unit format from t/m3 to t m-3 or tons m-3. Apply the same change to the all units throughout the manuscript.

Line 51: the mentioned values for “water holding capacity” are most likely reporting the gravimetric water content. Because water holding capacity should have a volumetric unit (e.g., cm3 cm-3) not gravimetric (g kg-1). I would review the related literature and revise the text accordingly.

Line 54: it is not the number of pores that is affected, instead the pore volume changes.

Line 55:  replace “pores” with “macropores”.

Lines 55-57: preferential flow pathways although increase water percolation, but negatively affect the matrix flow by rapid and concentrated transition of water and solute. So, they bring no benefit to water storage capacity. Please revise the paragraph.

Line 58: “humus” has a completely different definition. Please replace “humus” with “organic matter”.

Line 61-62: authors have frequently used the term “water infiltration” for “water transmission” or “water percolation”. The presence of standing vegetation is known to increase water infiltration, but as the focus of this manuscript is on the effect of root system on water dynamics, the water flux in subsurface should be named differently. Please note that the infiltration is the process of entering water into the soil, the water flux afterwards is usually called “transmission” or “percolation”. Please revise the entire manuscript accordingly.

Lines 60-62: the wording is unclear. Please make a revision.

Line 64: please consider replacing “soil porosity” with “soil macroporosity”.

Line 69: delete the “movement” at the end of sentence as its similar to flow.

Line 70-71: how a dead root can survive in a natural environment for more than 5 years without decomposition?! Please clarify.

Lines 71-84: this entire part is very confusing. Objectives more sounds like a shortened version of material and method. There is no clear statement on what issue is going to be addressed through this study. There is also no hypothesis as the objectives are missing.

Material and Methods

Lines 88-133: the entire Material and Methods section clearly shows why authors could not have a clear objective. Researchers doing experimental studies in environmental and agricultural sciences, either study soil in natural environment or try to resemble the natural environment in laboratory as much as possible (e.g., by collecting intact soil samples). That’s because their objectives are linked to an issue that exist in natural environment. Any discrepancy between the soil system that is studied in the lab and the one that exist in the nature would lead to useful set of information to deal with that specific issue.

Reviewer 2 Report

The subject of manuscript is very interesting. The manuscript investigated the influence of living and dead roots of poplar on water distribution and water flow movement in soil. The results showed that permeability and water storage capacity of soil could be improved by living and dead roots. The manuscript presents figures and tables which are supported the findings and conclusions of this study. However, the authors should be clarified the scientific benefit of knowing the water distribution and water movement of dead and living roots, the type of soil sample utilized in their study, and the values of hydraulic conductivity in their experiments. Below are some specific comments could improve the manuscript:

Lines 51-52: Adjusted the statement “the porosity and water holding capacity of soils increased from 41.2% to 46.3% to 4.3 g/kg to 4.8 g/kg, respectively” to “the porosity and water holding capacity of soils increased from 41.2% to 46.3% and from 4.3 g/kg to 4.8 g/kg, respectively”. Line 57: Delete the reference name (Lamandé et al., 2011) and replace it by the number of the reference [25] only. Lines 70-84: What is the scientific benefit of knowing the water distribution and water movement of dead and living roots? This should be clearly stated by authors before the objective of this study. Lines 76: Why the authors used this kind of plant (poplar)? What is the benefit of this plant? Why we need to know the water distribution and water movement of dead and living roots of poplar? This is should be clearly stated in the introduction before presented the objective of the study. Lines 96-102: What is the texture of soil samples? The percentage of sand, silt, and clay of soils should be stated. Did the authors use more than one soil type? The soils should be analyzed according to ASTM standards Figure 1: Point to the location of gravels and soil sample. Line 110: The authors used the word “And” more than 15 times in the beginning of sentences though out the manuscript. This is not appropriate.       Lines 119-127: During the experiments, Is the soil reached the saturation level? This is should be clarified by the authors. How the authors knowing the saturated level of soil sample during their experiments? Lines 127-128: Usually, Darcy law is used to estimate the water flux through a porous media. The equations 1 through 3 did not use Darcy law to estimate the water flux? Is Darcy law applicable for the experiments? Did the authors establish the soil permeability K throughout their experiments? Lines 143-144: How the authors ensure that the soil was not saturated? Did the authors use specific instrument to measure the saturation level of soil sample during the experiments? Lines 173-189: Why the authors did not establish the hydraulic conductivity K in their study?